# Effectiveness of dyadic interventions in improving outcomes for adults with multiple long-term conditions and/or frailty and their informal carers: A systematic review protocol

Stella Arakelyan[1]*, Michaela Gilarova[1], Mozhu Ding[2], Atul Jaiswal[3,4], Atul Anand[5], Karin Modig[2], Maria Kristiansen[6], Daniela Lillekroken[7], Susan D. Shenkin[1,8], Jill Manthorpe[9], Bruce Guthrie[1]

1 Advanced Care Research Centre, Usher Institute, University of Edinburgh, Edinburgh, United Kingdom, 2 Unit of Epidemiology, Institute of Environmental Medicine, Karolinska Institutet, Stockholm, Sweden, 3 Centre of Excellence in Frailty-Informed CareTM, Perley Health, Ottawa, Canada, 4 Interdisciplinary School of Health Sciences, Faculty of Health Sciences, University of Ottawa, Ottawa, Canada, 5 Centre for Cardiovascular Science, Chancellor's Building, University of Edinburgh, Edinburgh, United Kingdom, 6 Department of Public Health, University of Copenhagen, Copenhagen, Denmark, 7 Department of Nursing and Health Promotion, Faculty of Health Sciences, Oslo Metropolitan University, Oslo, Norway, 8 Ageing and Health, Usher Institute, University of Edinburgh, Edinburgh, United Kingdom, 9 NIHR Policy Research Unit in Health and Social Care Workforce, King's College London, London, United Kingdom

* stella.arakelyan@ed.ac.uk

## Abstract

### Aim

To synthesise current evidence on the effectiveness of dyadic (pair-based) interventions in improving outcomes for adults with multiple long-term conditions (MLTC) and/or frailty (aged ≥55 years) and their informal carers.

### Methods

The review protocol followed the Preferred Reporting Items for Systematic Review and Meta-Analysis Protocols (PRISMA-P) guidelines, with the protocol registered with PROSPERO (CRD420251144604). MEDLINE, Embase, PsycINFO, CINAHL Plus, CENTRAL, ClinicalTrials.gov will be searched for experimental and quasi-experimental studies examining the effectiveness of community-based dyadic interventions for adults with MLTC (≥2 long-term conditions within an individual) and/or frailty (aged ≥55 years) and their informal carers (spouses/partners, other family members or relatives) published since 2010 and up to September 2025. Dyadic interventions will be defined as pair-based interventions that directly involve informal carers and care recipient adults with MLTC and/or frailty using various techniques targeted at carers, care recipients, or both to change outcomes for at least one member of the carer/care recipient pair (or dyad). Database searches will be followed by a manual search of the reference lists of included studies and lists of papers citing

**Data availability statement:** As this manuscript is a systematic review protocol, there are no primary data associated with it. Accordingly, a data availability statement cannot be provided, as no datasets have yet been generated.

**Funding:** This work is funded by the Legal & General Group (research grant to establish the independent Advanced Care Research Centre at the University of Edinburgh) and Vivensa Foundation (ECRF24\4). The funders have no role in the conduct of this review, interpretation or the decision to submit for publication. The views expressed are those of the authors and not necessarily those of Legal & General or Vivensa Foundation.

**Competing interests:** The authors have declared that no competing interests exist. Although the Legal & General Group is a commercial funder, this does not alter their adherence to PLOS ONE policies on sharing data and materials.

included studies in order to identify additional studies. Two reviewers will independently screen titles and abstracts against the selection criteria and independently screen full texts using Covidence software. Methodological quality will be assessed using the Cochrane Risk of Bias (RoB) 2.0 tool for experimental studies and the Risk Of Bias In Non-randomised Studies of Interventions (ROBINS-I) tool for quasi-experimental studies. Synthesis of evidence will be quantitative. If meta-analysis is not possible, we will follow Cochrane recommendations for quantitative Synthesis Without Meta-Analysis (SWiM) guidance.

## Conclusion

The findings will address the evidence gap in dyadic implementation research in later life and help inform clinical decision-making, policy development and program planning for adults with MLTC and/or frailty and their carers, particularly in primary care and other community health settings.

## Introduction

The prevalence and complexity of multiple long-term conditions (MLTC or multi-morbidity) are increasing rapidly, especially among older people and those who are socio-economically deprived [1–4]. MLTC challenge health and social care systems as their presence is associated with increased dependency and complexity of care needs, higher health and care use and costs, poorer quality of life and mental health problems [5,6]. MLTC and frailty (decline in physiological reserves, contributing to increased vulnerability to adverse health outcomes) are interrelated, exacerbating the onset and severity of the other [7,8]. Over 16% of people with MLTC are frail, and over 70% of people who are frail live with MLTC [7]. Most people with MLTC and frailty live in community settings, where many rely on informal care and support from immediate family, relatives and friends to maximise quality of life and maintain independence. In 2022−23, an estimated 8% of the United Kingdom (UK) population (5.2 million people) were providing informal care, with a third of them aged ≥55 years [9]. Such care, which commonly includes support with activities of daily living, e.g., eating, dressing, grooming, bathing, shopping, attending appointments, managing and administering medications, often stems from love, respect, or duty, bringing positive benefits such as strengthened bonds and personal fulfilment. However, it frequently comes at a heavy cost to carers, especially to older spouse or partner carers who often live with long-term conditions themselves and have unmet support needs [10].

Care is a dynamic and mostly reciprocal relationship where the care recipient and informal carer both influence and are reliant on one another in different support domains [11–13]. Dyadic theoretical models, such as the Actor-Partner Interdependence Model [14] and the Theory of Dyadic Illness Management [15], provide a theoretical basis for understanding the reciprocal dynamics within care relationships by positing that an individual's traits and behaviours not only shape their own experiences and outcomes but also have effects on their care partner and vice versa. This

bidirectional and mutual influence underpins the rationale for dyadic or pair-based interventions, actively engaging carers and care recipients in interventions simultaneously to improve outcomes for one or both members of the dyad [16]. Given that poor experience of providing care for adults with MLTC and/or frailty may negatively affect care experiences and outcomes for both informal carers and care recipient adults with MLTC and/or frailty, there is a potential for dyad-tailored support interventions to achieve more meaningful and sustained improvements in dyadic outcomes. Previous reviews examining the effects of dyadic interventions on health outcomes have focused predominantly on informal carers and care recipient adults with a single disease (e.g., cancer, dementia) [17,18] or a mode of delivery of dyadic interventions (e.g., eHealth) [19]. The evidence of the benefits of dyadic interventions in the context of MLTC and frailty is lacking. It is also unclear if the effects of dyadic interventions differ according to informal carer characteristics, such as between co-resident older carers (spouses or partners), who typically provide higher-intensity care while managing their own long-term conditions [10] and adult family members, relatives or friends, who may or may not be co-resident and often provide lower-intensity support.

The aim of this review is to systematically synthesise current evidence on the effectiveness of dyadic interventions in improving outcomes for adults with MLTC and/or frailty (aged ≥55 years) and their informal carers. Specific research questions sought by the review are:

- What is the effectiveness of community dyadic interventions in improving outcomes for adults MLTC and/or frailty aged ≥55 years and their co-resident older informal carers (spouses or partners) aged ≥55 years?

- What is the effectiveness of community dyadic interventions in improving outcomes for adults MLTC and/or frailty (aged ≥55 years) and their informal carers (adult family members, relatives, or friends)?

## Materials and methods

The review protocol followed the Preferred Reporting Items for Systematic Review and Meta-Analysis Protocols (PRISMA-P) guidelines [20]. The protocol was registered with PROSPERO (CRD420251144604).

### Search strategy

Systematic searches will be performed of MEDLINE (Ovid), Embase (Ovid), PsycINFO (Ovid), CINAHL Plus (EBSCO), CENTRAL, and ClinicalTrials.gov for peer-reviewed literature published since 2010 and up to September 2025, with no language restrictions applied. The date limit is used to capture relatively recent and relevant intervention studies. The search strategy will apply subject terms and keywords relating to the target population and intervention, with the finalised search strategy being consulted with an information specialist and tailored to each database. A search strategy for MEDLINE (Ovid) is provided (S1 Table). In addition, we will manually search the reference lists of included studies and lists of papers citing included studies for other eligible studies.

### Population

We will include studies focusing on community-dwelling adults aged ≥55 years (or where the mean age ≥ 60 years) with MLTC and/or frailty and their informal carers (co-resident spouses/partners or other family members, relatives, friends). MLTC will be operationalised based on the NICE guideline definition as the presence of ≥2 long-term health conditions in an individual [21]. Long-term conditions will be defined based on the World Health Organisation's definition as persistent health problems needing ongoing management over a period of years or decades [22]. Frailty will be defined using either the frailty phenotype, the cumulative deficits model, or a validated frailty index or measurement tool. Community-dwelling will be defined as living independently at home (including in assisted living, extra-care and sheltered housing but excluding care/nursing home residents), regardless of the need for care assistance. An informal carer will be defined as an adult (a spouse, partner or other family member, relative or friend aged) who lives with or without a long-term condition(s) and

provides support with activities of daily living (eating, dressing, grooming, bathing, shopping, cooking, attending appointments, managing and administering medications, managing finances) to an adult with MLTC and/or frailty aged ≥55 years. We will exclude studies where care recipient adults with MLTC and/or frailty receive end-of-life or palliative care, receive support from paid carers only, have a single long-term condition or symptom, or those focusing on care recipient adults who all have a particular long-term condition or symptom (such as cancer, stroke, or dementia) with other comorbidities.

### Intervention

We will include studies that evaluate the effectiveness of dyadic interventions in the community setting (delivered at home, in other community settings, in primary care, or in ambulatory care/outpatient settings). There is no universally accepted and consistently applied definition of dyadic interventions. We will define dyadic interventions as pair-based interventions that directly involve informal carers and care recipient adults with MLTC and/or frailty using various techniques targeted at carers, care recipients or both to change outcomes for at least one member of the carer/care recipient pair (or dyad) [16].

### Comparators

We will consider studies reporting on any type of comparator intervention, including standard or usual care in the setting the intervention was implemented. Studies in which the comparator intervention involves minor enhancements to standard care, such as distribution of informational materials or training of carers or dyads, will also be eligible for inclusion if additional components are clearly stated and described.

### Types of studies

We will include experimental and quasi-experimental study designs, such as randomised controlled trials, non-randomised controlled trials, controlled before-and-after studies, and interrupted time series study designs. These are strong study designs acceptable for the evaluation of the effectiveness of health interventions by the Cochrane Effective Practice and Organisation of Care (EPOC) group criteria [23]. We will exclude studies reporting on observational study designs (e.g., case series, individual case reports, descriptive cross-sectional studies, case-control studies, cohort studies) and pharmacological studies. We will also exclude conference proceedings, abstracts, any type of reviews, and protocols.

### Outcomes

We will include studies reporting on quality of life (QoL), health and social care usage, including primary care usage (≥1 during follow up), emergency care attendance (≥1 during follow-up), hospital admission (≥1 during follow-up), social care usage (≥1 during follow up), caregiving burden and care competence, social participation or exclusion, mortality and other frequently reported outcomes measured by validated instruments or any clinically meaningful metrics. QoL could be measured by validated self-reported outcome instruments, such as the Short Form (SF)-12, SF-36, EQ-5D-3L, EQ-5D-5L. Caregiving or carer burden and care competence could be measured using tools capturing different aspects of carers' experience (e.g., emotional, physical, social, and financial impacts) and care competence (knowledge, skills, and attitudes) as covered in the Zarit Burden Interview, Caregiver Strain Index, Caregiver Reaction Assessment, Relative Stress Scale, Family Caregiver Task Inventory or Caregiver Self-Assessment Questionnaire. Social participation or exclusion may be measured using self-reported items describing frequency, quality, and diversity of involvement in social activities and community, such as Social Participation Questionnaire (SPQ), Late-Life Function and Disability Instrument (LLFDI), UCLA Loneliness Scale, Lubben Social Network Scale (LSNS)-6. This list is not exhaustive, and other validated measures of outcomes will also be considered. The review selection criteria are detailed in Table 1.

**Table 1. Study selection criteria.**

| Domain | Inclusion criteria | Exclusion criteria |
|---|---|---|
| **Publication type** | Experimental and quasi-experimental studies, grey literature (trial registers) | Conference proceedings, abstracts, any type of reviews and study protocols |
| **Publication timeline** | Published since 2010 | |
| **Population** | | |
| **Care recipients** | Community-dwelling (living independently at home) adults with MLTC (≥2 long-term health conditions in an individual) and/or frailty aged ≥55 years (mean age ≥ 60 years) | Community-dwelling adults who are receiving end-of-life or palliative care<br>Community-dwelling adults with a particular long-term condition (cancer, stroke, dementia)<br>Community-dwelling adults with MLTC and/or frailty with no family/friend care<br>Community-dwelling adults with MLTC and/or frailty receiving only paid care and support |
| **Informal carers** | An adult (spouse, partner or other family member, relative or friend) who lives with or without a long-term condition(s) and provides support with activities of daily living to a community-dwelling adult with MLTC and/or frailty aged ≥55 years | |
| **Intervention** | Dyadic or pair-based interventions that directly involve informal carers and care recipient adults with MLTC and/or frailty using various techniques targeted at carers, care recipients, or both to change outcomes for at least one member of the carer/care recipient pair (or dyad) | |
| **Comparator** | Context-specific standard care or usual care and enhanced standard care if additional components are clearly stated and described | |
| **Outcomes** | Quality of life, emergency care attendance, hospital admission, primary care usage, social care usage, caregiving/carer burden and care competence, social participation or exclusion, mortality | |
| **Context** | Community setting of intervention (at home, in other community settings, in primary care, or in ambulatory care/outpatient settings) | Hospice, care/nursing home, end-of-life care settings |
| **Study designs** | Randomised controlled trials, non-randomised controlled trials, controlled before-after studies, interrupted time series | Case series, individual case reports, descriptive cross-sectional studies, case-control, and cohort studies |

## Data extraction

We will import retrieved records to EndNote v20.3 (Clarivate Analytics, PA, USA) for de-duplication. RIS files will be transported to Covidence (Veritas Health Innovation, Melbourne, Australia) for data management, with initial title/abstract screening and further full-text screening done by two independent reviewers. Discrepancies between reviewers will be resolved by discussion and, if necessary, the involvement of a third reviewer. We will extract data using a structured data extraction tool on (1) study characteristics (title, first author, country, year of publication, objective); (2) included populations (age, gender, other socio-demographic details, number and type of conditions, definitions, and measures used); (3) search strategy; (4) dyadic interventions (types of interventions, country in which interventions were tested, intervention components, who delivered, who was the target(s), type of controls, total sample sizes); (5) setting; (6) health and care outcomes (measures used, measured for whom) and results. Authors of original studies will be contacted for incomplete or missing data.

## Quality appraisal

Methodological quality will be assessed by two reviewers independently using tools tailored to study design. For RCTs, we will use the Cochrane Risk of Bias (RoB) 2.0 tool [24], which evaluates potential sources of bias across five domains: bias arising from the randomisation process, deviations from intended interventions, missing outcome data, measurement

of the outcome, and selection of the reported result. Each domain will be rated as having a 'low risk of bias', 'some concerns', or a 'high risk of bias', with an overall risk of bias based on the judgments across all five domains. For quasi-experimental studies, we will use the Risk Of Bias In Non-randomised Studies of Interventions (ROBINS-I) tool [25]. This tool assesses bias across seven domains, including confounding, selection of participants, classification of interventions, deviations from intended interventions, missing data, measurement of outcomes, and selection of reported results. Each domain will be rated on a spectrum from low to critical risk, with the overall risk of bias determined by the highest risk level across all domains. Discrepancies between reviewers' ratings will be resolved by discussion, and if necessary, authors of original studies will be contacted for additional information.

## Analysis

If appropriate, meta-analyses will be conducted by pooling risk ratios for dichotomous outcomes (e.g., hospital admission, mortality), or standardised mean differences (SMDs) with 95% confidence intervals for continuous outcomes (quality of life, social participation or exclusion). Heterogeneity will be assessed using the $I^2$ statistic and the omnibus homogeneity test (Q) using metrics: 0%−40% (low heterogeneity); 30%−60% (moderate heterogeneity); 50%−90% (substantial heterogeneity); and 75%−100% (considerable heterogeneity). A fixed-effects model will be used if the heterogeneity is low; otherwise, we will use a random-effects model and explore potential causes of heterogeneity using subgroup analysis. For RCTs with multiple periods of follow-up, the aim will be to meta-analyse at different time-points (e.g., end of the intervention period and varying durations of follow-up after the end of active intervention), depending on what is possible with reported data. If possible, subgroup analyses will be conducted according to dyadic intervention types (e.g., educational, psychosocial, digital) and outcomes. Publication bias will be assessed using funnel plots and Egger's test, with statistical significance set at $p < 0.05$. For data where meta-analysis is not appropriate, synthesis will be quantitative where possible, following Cochrane recommendations for quantitative Synthesis Without Meta-Analysis (SWiM) guidance [26]. The Grading of Recommendations Assessment, Development and Evaluation (GRADE) approach will be used for the assessment of the overall quality of evidence.

## Ethical considerations

Ethical approval is not required for this systematic review and its protocol as it involves secondary analysis of previously published data.

## The status and timeline of the study

As of 22 September 2025, systematic searches across six databases have been completed. Screening is in progress and is expected to conclude by the end of October 2025. Quality appraisal and data extraction are anticipated to be completed by December 2025, with the full systematic review and reporting of results expected within the following 12 months.

## Discussion

The growing prevalence and complexity of MLTC present an urgent challenge for health and social care systems worldwide. While care and support provided to community-dwelling adults with MLTC and/or frailty is often delivered within reciprocal dyadic relationships, existing evidence has largely overlooked this relational dimension, focusing instead on single-disease contexts or specific modes of intervention delivery. To achieve more meaningful and sustained improvements in dyadic outcomes, it is important to provide a systematic synthesis of the effects of various dyadic interventions targeting adults with MLTC and/or frailty and their informal carers, and map the features of community settings in which these interventions were developed and evaluated in relation to various outcomes. By applying a dyadic lens, this systematic review will address this evidence gap and advance understanding of how community-based care and support services

could be re-designed, tailored and implemented to improve outcomes for both adults MLTC and/or frailty and their informal carers, supporting them to live well and longer in their communities.

Findings will have implications for research, policy, and practice. For researchers, it will provide clear evidence of benefit of dyadic interventions in context of MLTC and frailty and identify gaps in implementation research in later life, such as the types of dyadic interventions that have been most rigorously evaluated, outcomes frequently measured and those overlooked, and the methodological limitations that require further exploration. For service providers and policy-makers, the review will provide evidence-based insights into the types of community-based dyadic interventions that have the potential to reduce unmet support needs and enhance care experiences and outcomes for people with MLTC and/or frailty and their carers. To help bridge the gap between research and practice, the review findings will be disseminated not only through academic publications but also, in collaboration with key stakeholders, co-produced as an "effective action policy brief" that translates evidence into practical, actionable guidance. This guidance can inform the co-development and implementation of dyadic interventions that are effective, acceptable, feasible, and tailored to local needs and contexts.

## Conclusion

Review findings will address the evidence gap in dyadic implementation research in later life and help inform clinical decision-making, policy development and program planning for adults with MLTC and/or frailty and their informal carers, particularly in primary care and other community health settings.

## Supporting information

**S1 Table. Search strategy for MEDLINE (Ovid interface).**
(DOCX)

## Author contributions

**Conceptualization:** Stella Arakelyan, Atul Anand, Jill Manthorpe, Bruce Guthrie.

**Funding acquisition:** Stella Arakelyan.

**Methodology:** Stella Arakelyan, Michaela Gilarova, Mozhu Ding, Atul Jaiswal, Atul Anand, Karin Modig, Maria Kristiansen, Daniela Lillekroken, Susan D. Shenkin, Jill Manthorpe, Bruce Guthrie.

**Project administration:** Stella Arakelyan.

**Writing – original draft:** Stella Arakelyan, Bruce Guthrie.

**Writing – review & editing:** Stella Arakelyan, Michaela Gilarova, Mozhu Ding, Atul Jaiswal, Atul Anand, Karin Modig, Maria Kristiansen, Daniela Lillekroken, Susan D. Shenkin, Jill Manthorpe, Bruce Guthrie.

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
