## [Decision Letter · Decision Letter 0]

22 Dec 2025

Dear Dr. Arakelyan,

Thank you for submitting your manuscript to PLOS ONE. After careful consideration, we feel that it has merit but does not fully meet PLOS ONE’s publication criteria as it currently stands. Therefore, we invite you to submit a revised version of the manuscript that addresses the points raised during the review process.

We look forward to receiving your revised manuscript.

Kind regards,

Robbert Huijsman, PhD

Academic Editor

PLOS One

Journal Requirements:

[This work is funded by the Legal & General Group (research grant to establish the independent Advanced Care Research Centre at the University of Edinburgh) and Vivensa Foundation (ECRF24\4). The funders have no role in the conduct of this review, interpretation or the decision to submit for publication. The views expressed are those of the authors and not necessarily those of Legal & General or Vivensa Foundation.]

We note that you received funding from a commercial source: Legal & General Group

Additional Editor Comments:

Thank you for tou clear protocol paper.

The reviewers give you some minor points for revision.

Reviewers' comments:

Reviewer's Responses to Questions

**Comments to the Author**

1. Does the manuscript provide a valid rationale for the proposed study, with clearly identified and justified research questions?

Reviewer #1: Partly

2. Is the protocol technically sound and planned in a manner that will lead to a meaningful outcome and allow testing the stated hypotheses?

Reviewer #1: Yes

3. Is the methodology feasible and described in sufficient detail to allow the work to be replicable?

Reviewer #1: Yes

4. Have the authors described where all data underlying the findings will be made available when the study is complete?

Reviewer #1: Yes

5. Is the manuscript presented in an intelligible fashion and written in standard English?

Reviewer #1: Yes

You may also provide optional suggestions and comments to authors that they might find helpful in planning their study.

Reviewer #1: Dear Dr. Arakelyan,

Thank you for your manuscript and the interesting topic presented. This manuscript addresses an important research area and presents a well-structured protocol. I have a few concerns regarding the methodology and focus of this protocol.

1) Methods: In the text, you claim to include articles from 2010 onward. However, in Table 1. Study selection criteria, the search inclusion is from 2016. Please clarify the inclusion criteria and remain consistent.

2) Please provide clearer examples of comparators.

3) The two-pronged aim is focusing on a) co-resident older informal caregiver and b) informal caregiver (friends, family). The value of the delineation is unclear in the manuscript and should be further elaborated.

4) The protocol has not clarified how a review of intervention effectiveness can inform the re-design, tailoring, and implementation of health services. There is a gap between effectiveness research and implementation research that the author should clearly address.

**Do you want your identity to be public for this peer review?** For information about this choice, including consent withdrawal, please see our Privacy Policy

Reviewer #1: No

---

## [Author Response · Author response to Decision Letter 1]

5 Jan 2026

We appreciate the helpful and constructive comments provided by editorial team and Reviewer 1 which we have addressed in full.

Comment 1: Methods: In the text, you claim to include articles from 2010 onward. However, in Table 1. Study selection criteria, the search inclusion is from 2016. Please clarify the inclusion criteria and remain consistent.

Response 1: Thank you for highlighting this. We confirm that articles published from 2010 onward were eligible for inclusion. Table 1 has been corrected to reflect this.

Comment 2: Please provide clearer examples of comparators.

Response 2: We consider for inclusion any comparator used in the original studies, including standard or usual care and modified usual care if additional components are clearly stated and described. We have revised the manuscript to clarify and added examples (please see the revised text below).

We will consider studies reporting on any type of comparator intervention, including standard or usual care in the setting the intervention was implemented. Studies in which the comparator intervention involves minor enhancements to standard care, such as distribution of informational materials or training of carers or dyads, will also be eligible for inclusion if additional components are clearly stated and described.

Comment 3: The two-pronged aim is focusing on a) co-resident older informal caregiver and b) informal caregiver (friends, family). The value of the delineation is unclear in the manuscript and should be further elaborated.

Response 3: We described in the introduction (lines 82-83) that informal carers often experience negative effects of caregiving, and these effects are more prominent in older spouse or partner carer who commonly provide more intensive support and live with own long-term conditions and have many unmet support needs. In contrast, adult family members, relatives, or friends may or may not be co-resident and typically provide care of lower intensity and shorter duration. Distinguishing these groups provides a more nuanced understanding of caregiving contexts, needs, and intervention effects. We have revised introduction to better articulate this delineation (please see the revised text below).

It is also unclear if the effects of dyadic interventions differ according to informal carer characteristics, such as between co-resident older carers (spouses or partners), who typically provide higher-intensity care while managing their own long-term conditions [10] and adult family members, relatives or friends, who may or may not be co-resident and often provide lower intensity support.

Comment 4: The protocol has not clarified how a review of intervention effectiveness can inform the re-design, tailoring, and implementation of health services. There is a gap between effectiveness research and implementation research that the author should clearly address.

Response 4: We agree that the gap between evidence and practice remains a challenge, and progress in addressing it has been slow. Contributing factors include unclear communication of research findings to service providers and policymakers, as well as limited financial and political support. While researchers alone cannot resolve funding constraints or ensure political commitment, we will work collaboratively with stakeholders to co-produce an “effective action policy brief” that translates evidence into practical, actionable guidance. This guidance will support the co-development and implementation of dyadic interventions that are effective, acceptable, feasible, and tailored to local contexts. The final paragraph of the discussion has been revised to reflect this (please see below).

For service providers and policymakers, the review will provide evidence-based insights into the types of community-based dyadic interventions that have the potential to reduce unmet support needs and enhance care experiences and outcomes for people with MLTC and/or frailty and their carers. To help bridge the gap between research and practice, the review findings will be disseminated not only through academic publications but also, in collaboration with key stakeholders, co-produced as an “effective action policy brief” that translates evidence into practical, actionable guidance. This guidance can inform the co-development and implementation of dyadic interventions that are effective, acceptable, feasible, and tailored to local needs and contexts.

---

## [Editor Report · Decision Letter 1]

15 Jan 2026

Effectiveness of dyadic interventions in improving outcomes for adults with multiple long-term conditions and/or frailty and their informal carers: A systematic review protocol

PONE-D-25-50837R1

Dear Dr. Arakelyan,

We’re pleased to inform you that your manuscript has been judged scientifically suitable for publication and will be formally accepted for publication once it meets all outstanding technical requirements.

Kind regards,

Robbert Huijsman, PhD

Academic Editor

PLOS One

Additional Editor Comments (optional):

Thanks to the authors for their revision and carefull responses to the reviewers.

There are no further comments, the revision is satisfactory.
---

## [Editor Report · Acceptance letter]

PONE-D-25-50837R1

PLOS One

Dear Dr. Arakelyan,

I'm pleased to inform you that your manuscript has been deemed suitable for publication in PLOS One. Congratulations! Your manuscript is now being handed over to our production team.

Kind regards,

on behalf of

Professor Robbert Huijsman

Academic Editor

PLOS One